# Prediction of Cardiorespiratory Fitness in Czech Adults: Normative Values and Association with Cardiometabolic Health

**DOI:** 10.3390/ijerph181910251

**Published:** 2021-09-29

**Authors:** Geraldo A. Maranhao Neto, Iuliia Pavlovska, Anna Polcrova, Jeffrey I. Mechanick, Maria M. Infante-Garcia, Jose Pantaleón Hernandez, Miguel A. Araujo, Ramfis Nieto-Martinez, Juan P. Gonzalez-Rivas

**Affiliations:** 1International Clinical Research Center (ICRC), St Anne’s University Hospital (FNUSA), 656 92 Brno, Czech Republic; iuliia.pavlovska@fnusa.cz (I.P.); anna.polcrova@fnusa.cz (A.P.); maria.garcia@fnusa.cz (M.M.I.-G.); juan.gonzalez@fnusa.cz (J.P.G.-R.); 2Department of Public Health, Faculty of Medicine, Masaryk University, 656 91 Brno, Czech Republic; 3Research Centre for Toxic Compounds in the Environment (RECETOX), Masaryk University, 656 91 Brno, Czech Republic; 4The Marie-Josée and Henry R. Kravis Center for Cardiovascular Health at Mount Sinai Heart, Icahn School of Medicine at Mount Sinai, New York, NY 10029, USA; jeffreymechanick@gmail.com; 5Division of Endocrinology, Diabetes and Bone Disease, Icahn School of Medicine at Mount Sinai, New York, NY 10029, USA; 6Foundation for Clinic, Public Health, and Epidemiology Research of Venezuela (FISPEVEN INC), Caracas 1060, Venezuela; nieto.ramfis@gmail.com; 7EDU Medicine and Health, Digital Education Holdings Ltd., KKR 1320 Kalkara, Malta; j_pantaleon@hotmail.com; 8Department of Physical Education, School of Education, University of Los Andes, Mérida 5101, Venezuela; migmaar@hotmail.com; 9Department of Global Health and Population. Harvard TH Chan School of Public Health, Harvard University, Boston, MA 02115, USA; 10LifeDoc Health, Memphis, TN 38119, USA

**Keywords:** cardiorespiratory fitness, cardiometabolic risk factors, population health, adult, middle aged

## Abstract

Cardiorespiratory fitness (CRF) is a strong independent predictor of morbidity and mortality. However, there is no recent information about the impact of CRF on cardiometabolic risk specifically in Central and Eastern Europe, which are characterized by different biological and social determinants of health. In this cross-sectional study normative CRF values were proposed and the association between CRF and cardiometabolic outcomes was evaluated in an adult Czechian population. In 2054 participants (54.6% females), median age 48 (IQR 19 years), the CRF was predicted from a non-exercise equation. Multivariable-adjusted logistic regressions were carried out to determine the associations. Higher CRF quartiles were associated with lower prevalence of hypertension, type 2 diabetes (T2D) and dyslipidemia. Comparing subjects within the lowest CRF, we see that those within the highest CRF had decreased chances of hypertension (odds ratio (OR) = 0.36; 95% CI: 0.22–0.60); T2D (OR = 0.16; 0.05–0.47), low HDL-c (OR = 0.32; 0.17–0.60), high low-density lipoprotein (OR = 0.33; 0.21–0.53), high triglycerides (OR = 0.13; 0.07–0.81), and high cholesterol (OR = 0.44; 0.29–0.69). There was an inverse association between CRF and cardiometabolic outcomes, supporting the adoption of a non-exercise method to estimate CRF of the Czech population. Therefore, more accurate cardiometabolic studies can be performed incorporating the valuable CRF metric.

## 1. Introduction

Cardiovascular disease (CVD) is the leading cause of mortality in Europe, accounting for 45% of all deaths [1]. Higher age, male sex, hypertension, type 2 diabetes (T2D), dyslipidemia, obesity, low physical activity, and smoking are the traditional risk factors for CVD [2]. Over the last three decades, cardiorespiratory fitness (CRF) has emerged as a strong independent predictor of all-cause and disease-specific mortality [3]. CRF is considered to be a potentially stronger predictor of mortality than traditional risk factors [4]. CRF is an important marker of cardiovascular health and provides a measure of the body’s ability to transport, absorb, and utilize oxygen to energy transfer to the muscles during physical activities [5]. Improvement in CRF is associated with a lower risk of all-cause [6] and CVD mortality [7]. This improvement favorably influences the health by reducing cardiometabolic risk characterized by dysglycemia, abnormal adiposity, hypertension, and dyslipidemia [8]; reducing adiposity, platelets adhesion, and aggregation; increasing skeletal muscle mass, insulin sensitivity, glucose disposal, and lipoprotein lipase function; and improving lipid profile [9].

Valid and representative reference values are critical for the interpretation of CRF, and when possible, data should be reported in the context of already published data from the same reference population [10]. However, there is limited information about the CRF in Central and Eastern European populations. Two studies are noteworthy: the Czechoslovakian study of 1071 subjects aged between 12 and 55 years [11], and the Lithuanian study of 168 adults, aged between 20 and 60 years [12]. The first study [11] used the data collected more than 40 years ago, which limited the current applicability of results, especially since concepts of health promotion and primordial/primary prevention only emerged in the former Czechoslovakia after 1990 [8]. In the second study [12], the sample was not population representative.

Western European countries have more representative and normative data [13,14,15,16,17], enabling subsequent studies on the associations among CRF, cardiometabolic risk factors, and clinical outcomes in adults [18,19,20,21,22,23,24,25,26,27]. However, there is no representative and normative data on CRF in Central and Eastern European populations, limiting relevant information on the health status of this population. The high costs of implementation and time spent on CRF testing may be a principal reason for this lack of information, especially in epidemiological studies [28]. A non-exercise prediction model is a feasible alternative to assess CRF without exercise testing. In this case, CRF is estimated instead through regression equations and based on easily available variables (e.g., body mass index (BMI), waist circumference (WC), resting heart rate (RHR), smoking profile, and physical activity (PA) level), showing good accuracy and predictive ability [29]. This alternative approach has been associated with health outcomes in a heterogeneous and population-representative sample [30].

The present paper aims to (a) propose normative CRF values of a sample of adults aged between 25 and 64 years old in Czechia, estimated through a non-exercise prediction model CRF; and (b) evaluate the association between estimated CRF and cardiometabolic outcomes.

## 2. Materials and Methods

### 2.1. Study Design and Population

The study design was described previously [31]. In brief, the Kardiovize study is a cross-sectional, random evaluation of adults between 25 and 64 years old of Brno, Czechia. Brno is the second-largest city in Czechia, with 373,327 residents. Eligibility criteria included permanent residence in the city and registration (required by the law) with any of the 5 state-run health insurance companies

### 2.2. Sampling

Survey sampling was performed in January 2013 with technical assistance from the largest (state-run) health insurance company using the registries of all health insurance companies. Registration with a health insurance company is mandatory in the Czech Republic. No a priori calculation of the sample size was performed. A random sample of 3300 persons stratified by age and gender was adjusted for a response rate of 64.4% (as projected from the Czech post-MONICA study). Health insurance companies mailed invitation letters with a description of the study, ensuring confidentiality. Similar to the post-MONICA study, Kardiovize targeted 1% of the adult urban population between 25 and 64 years old. Because the sample size was not reached, a second random sampling was performed, following the same methodology, with 3077 mailed invitations. Based on the two samplings with a total of 6377 randomly selected invitees, the overall response rate was 33.9% [31]. No information on non-respondents was available. For this analysis, subjects with type 1 diabetes were excluded. A total of 2160 individuals signed informed consent to participate and were enrolled, which was large enough to ensure the representativeness of various sociodemographic strata in the sample.

### 2.3. Data Collection

The questionnaires included demographics (age, education, and marital status), socioeconomic status, cardiovascular risk behaviors (smoking status, nutrition, alcohol consumption, and physical activity), family and personal history, medications, hospitalizations, and mental health. Laboratory analyses were performed on 12-h fasting whole blood samples, using a Modular SWA P800 analyzer (Roche, Basel, Switzerland); total cholesterol, triglycerides, and glucose were analyzed by using the enzymatic colorimetric method (Roche Diagnostics GmbH, Mannheim, Germany); and high-density lipoprotein cholesterol (HDL-c) was analyzed with the homogeneous method for direct measurement without precipitation (Sekisui Medical, Hachimantai, Japan). Low-density lipoprotein cholesterol (LDL-c) level was calculated according to the Friedewald equation when triglyceride levels were below 4.5 mmol/L; if it was higher, LDL-c was analyzed by using the homogeneous method for direct measurement (Sekisui Medical, Hachimantai, Japan). Blood pressure (BP) was measured with the patient alone, using an automated office measurement device (BpTRU, model BPM 200; Bp TRU Medical Devices Ltd., Coquitlam, BC, Canada).

### 2.4. Variables Definition

Physical activity was assessed by using the International Questionnaire of Physical Activity (IPAQ), long version. Subjects were categorized as “active” if they had participated in vigorous physical activity 3 or more days per week, at least 20 min per day; moderate-intensity physical activity or walking 5 or more days, at least 30 min per day; or any combination of walking, moderate-intensity activities, or vigorous-intensity activities 5 or more days per week, achieving a minimum of at least 600 metabolic equivalent for task (MET)-min/week [32]. Subjects categorized as “insufficiently active” were those who did not perform in any of the activities above.

Marital status was categorized into living alone (including single, divorced, and widowed) or living in a couple (including married and other partnerships). Educational level was categorized as primary, secondary, and higher. Household income was expressed in Euros/month and categorized as low “<1200”, middle “1200–1800”, or high “>1800”. Smoking status was classified as “non-smokers” or “current smokers” (smoking daily or less than daily during the past year). Participants were categorized into “non-drinkers” (including abstainers and those who did not drink in the previous 12 months) and “drinkers”. Alcohol consumption was assessed by the reported alcohol intake of the last week, expressed in the number of standard drinks. One standard drink was assessed as a glass of wine, bottle of beer, or shot of spirits, each corresponding to approximately 10 g of ethanol.

### 2.5. Cardiometabolic Outcomes

Hypertension was defined as BP ≥ 140/90 mmHg, self-report of hypertension, or taking anti-hypertensive medications. Type 2 diabetes was defined as fasting blood glucose ≥ 7 mmol/L, self-report of T2D, or taking antidiabetic medications. Low HDL-cholesterol was defined as <1 mmol/L in men or <1.2 mmol/L in women; high LDL-cholesterol was defined as LDL ≥ 3 mmol/L or on lipid-lowering drugs, such as fibrates, nicotinic acid, and statins; hypertriglyceridemia was defined as triglycerides ≥ 1.7 mmol/L or on lipid-lowering drugs; hypercholesterolemia as total cholesterol ≥ 5.0 mmol/L or on lipid-lowering drugs [33].

### 2.6. Cardiorespiratory Fitness Estimation

The CRF was predicted from the non-exercise equation of Jackson et al. [34], including age, body mass index (BMI), waist circumference (WC), and resting heart rate (RHR) as continuous variables; and physical activity (PA) level and smoking status as dichotomous variables. The CRF values are expressed in MET units. The prediction equations were as follow:CRF (Men): 21.2870 + (age × 0.1654) − (age2 × 0.0023) − (BMI × 0.2318) − (WC × 0.0337) − (RHR× 0.0390) + (PA level × 0.6351) − (smoking status × 0.4263)
CRF (Women) = 14.7873 + (age × 0.1159) − (age2 × 0.0017) − (BMI × 0.1534) − (WC × 0.0088) − (RHR× 0.0364) + (PA level × 0.5987) − (smoking status × 0.2994)

### 2.7. Data Analysis

All statistical analyses were performed by using the STATA software (version 14.0, StataCorp, College Station, TX, USA). The Kolmogorov–Smirnov test was used to assess the normal distribution of variables BMI, waist circumference, and resting heart rate. Continuous variables were reported as the median and interquartile range (IQR) and compared by using the Mann–Whitney U test. Categorical variables were reported as frequency and percentage and compared using the Chi-squared or Fisher test. The Chi-squared test was used to assess the differences between variables across the cardiometabolic outcomes status and CRF categories. The CRF in METs was categorized in quartiles. The quartiles were sex- and age-specific. Crude and multivariable-adjusted logistic regressions were carried out to determine the association between CRF and the presence of cardiometabolic outcomes. Model 1 was unadjusted. Model 2 was adjusted by age, sex, BMI, educational level, smoking status, alcohol drinking, and diuretic and vasodilator use (in case of hypertension); antidiabetic medication use (in case of T2D), and lipid-lowering medication use (in case of dyslipidemia). Statistical significance was set at *p* < 0.05.

## 3. Results

### 3.1. Subjects’ Characteristics

In total, 2054 subjects were included (Figure 1); 45.4% were men, with a median age of 48.0 (IQR 19.0) years (Table 1). Men had a higher BMI and WC, and a higher prevalence of hypertension, T2D, high LDL-c, high cholesterol, high triglycerides, alcohol use, high education, and income levels than women. Women had a higher heart rate than men. Both sexes reported a high prevalence of sufficient physical activity (87.7% in men and 88.8% in women; *p* = 0.279) and similar prevalence of smoking (25.2% in men and 22.0% in women; *p* = 0.231).

### 3.2. Association of Cardiorespiratory Fitness and Cardiometabolic Risk Factors

The cutoff values in METs classified by age and sex to define the quartiles used are presented in Table 2. All the analyses considered the first quartile (Q1) as the lowest cutoff for CRF. Higher quartiles of CRF were associated with a lower prevalence of hypertension, T2D, and dyslipidemias (Table 3). Using logistic regression analysis, in the raw Model 1, we show a significant progressive reduction in the prevalence of cardiometabolic risk factors by each quartile and each MET estimated, except with high LDL-c and high cholesterol, where the difference was only observed with the highest quartile (Table 3). The fully adjusted Model 2 showed an independent association of CRF with all the assessed risk factors. Comparing with subjects with the lowest CRF (worse condition), we see that those with the highest CRF had a decreased chance of having hypertension by 64% (odds ratio (OR) = 0.36; 95% CI 0.22–0.60), T2D by 84% (OR = 0.16; 95% CI 0.05–0.47), low HLD-c by 68% (OR = 0.32; 95% CI 0.17–0.60), high LDL-c by 67% (OR = 0.33; 95% CI 0.21–0.53), high triglycerides by 87% (OR = 0.13; 95 CI 0.07–0.81), and high cholesterol by 56% (OR = 0.44; 95% CI 0.29–0.69) (Table 3).

## 4. Discussion

This is the first study aimed to describe the distribution of predicted CRF levels among men and women aged 25–64 years, divided into decades in the Czech population that can be used as a reference for future clinical studies (Table 2). The classification in quartiles was associated with several cardiometabolic outcomes. The main findings can be summarized as follows: after adjustment by age categories, sex, BMI, educational level, smoking, alcohol use, and specific medications, the prevalence of hypertension, T2D, and hypertriglyceridemia were inversely associated with higher CRF quartile. In addition, low HDL-c, high LDL-c, and hypercholesterolemia were inversely associated with quartiles three and four (Table 3).

Results from other studies are easily converted to METs (VO_2_ mL·kg^−1^·min^−1^/3.5 mL·kg^−1^·min^−1^), allowing for a comparison with the present findings. For instance, the mean results (METs) in the present study (men 30–39 years = 12.9, 40–49 years = 12.4, and 50–59 years = 11.1; women 30–39 years = 10.4, 40–49 years = 9.7, and 50–59 years = 9.0) are higher in each age classification than those established from the Czechoslovakian population in the 1970s: (men 30–39 years = 10.3, 40–49 years = 10.1, and 50–59 years = 9.3; women 30–39 years = 8.4, 40–49 years = 7.9, and 50–59 years = 7.2) [11]. As an illustration of how specific normative data can be useful, the values corresponding to the 25th, 50th, and 75th percentiles for a 45-year-old man in the present study were generally higher than values in German and United Kingdom registries, with values for a 45-year-old woman roughly the same for men (11.0, 12.3, and 13.1) and women (9.6, 10.1, and 10.5) in the present study; and compared with men (7.1, 10.8, and 13.7) and women (5.7, 8.6, and 11.7) in Germany, and men (8.9, 10.6, and 12.6) and women (5.7, 8.6, and 11.7) in the United Kingdom, respectively [17]. The comparison with other populations is important to highlight particularities of the studied sample, but also with the goal of including CRF as another risk factor under surveillance and creating CRF registry in different countries.

The relationship between CRF and cardiometabolic outcomes has different mechanisms. Aerobic activities are known to influence CRF levels and to prevent development of hypertension [35]. Blood pressure reduction with higher CRF levels appears to result from a decrease in total peripheral resistance, due to neurohumoral and structural vascular adaptations, decreased sympathetic tone, increased local vasodilatory responses, and favorable changes among endogenous vasoconstriction and vasodilation factors [36]. Better CRF allows performing aerobic exercises with greater intensity, duration, and frequency. This translates into higher caloric expenditure during and after exercise, decreasing the risk for abnormal adiposity [37]. In addition, a higher CRF is a cornerstone in the control and prevention of T2D, by improving insulin sensitivity and maintaining glycemic control [30,38]. Improvements in CRF have also been associated with better HDL functionality, including higher cholesterol efflux capacity. Antioxidative and anti-inflammatory HDL properties inhibit the oxidation of LDL. Moreover, HDL can take-up lipid peroxides, by-products of lipid oxidation, and transport them to the liver for excretion [39,40]. Increased HDL antioxidative capacity has been related to decreased cholesterol and triglyceride content [41].

The present results are consistent with previous literature using non-exercise models. For instance, the prevalence of hypertension in a sample from United States population (20–86 years) significantly decreased from the highest to lowest CRF tertiles (from 42.5% to 24.3% in men and 45.5 to 20.6 in women) [42] and it was also observed in British adults (35–70 years) (CRF quartiles from 32.9% to 8.9% in men and 37.2% to 7.2% in women) by Stamatakis et al. [43]. The same study showed a prevalence of T2D significantly decreasing across the CRF quartiles (from 7.0% to 0.9% in men and 5.8% to 0.4% in women), which was also observed in Brazilian subjects (20–59 years) (22.9% versus 0.9% in men and 12.2% versus 1.8% in women) [30]. The association between CRF and dyslipidemia is very well documented [44,45,46]. In fact, there are non-exercise models that include dyslipidemia as an independent variable [47]. In relation to the specific components of dyslipidemia (low HDL-c, high LDL-c, hypertriglyceridemia, and hypercholesterolemia), to the best of our knowledge, this is the first study showing associations between non-exercise CRF and each one of the components of dyslipidemia.

Epidemiological data recently suggested that having a better CRF significantly decreases the risk of hospitalization and dying from COVID-19 [48,49]. This decrease in risk can be mediated by an improvement in the cardiometabolic profile especially in individuals with obesity or overweight, insulin resistance, and diabetes [50]. Physical activity is a modifiable behavior that positively influences CRF level and can be a mitigation strategy against COVID-19 [49], especially in Czechia, where, in contrast to the decreasing burden and mortality related to CVD and certain cardiometabolic outcomes, the prevalence of dysglycemia and abnormal adiposity are increasing [8]. However, primary prevention programs in Czechia still require improvement. Concepts of health promotion and primary/primordial prevention only emerged in former Czechoslovakia after political change in 1990, and a system of public health control is still being established [8]. Therefore, the number of physically inactive adults has been increasing [51], and only recently have multi-morbidity care models [52] been applied [53,54]. A diligent focus on risk factors, including physical activity and fitness, is needed to slow down, stop, or even reverse cardiometabolic-based chronic disease progression [8,53].

Among the main criticisms of non-exercise models are the inability to account for inter-individual variability [55]. However, the capacity to classify the CRF levels [56] and to associate them with cardiometabolic variables makes the method very useful for epidemiological studies. This study is the first to present the associations of a non-exercise model of CRF assessment with a broad range of cardiovascular outcomes. The present study also demonstrates generalizability of the model in a different and representative population. The city of Brno, a single urban setting, would represent the urban population of Czechia [57]. Limitations of the analysis are related to the cross-sectional design, which does not allow a causal relationship and the inclusion of self-reported behaviors, especially physical activity level, which might bias results.

## 5. Conclusions

Age- and sex-specific CRF normative values were estimated from a non-exercise model in a representative sample of 2054 Czech citizens from 25 to 64 years old. The generated CRF quartiles showed an inverse association with different cardiometabolic outcomes and support the adoption of the non-exercise method to obtain relevant information on the health status of the Czech population. With these results, more accurate cardiometabolic studies can be performed that incorporate the valuable CRF metric.

## Figures and Tables

**Figure 1 ijerph-18-10251-f001:**
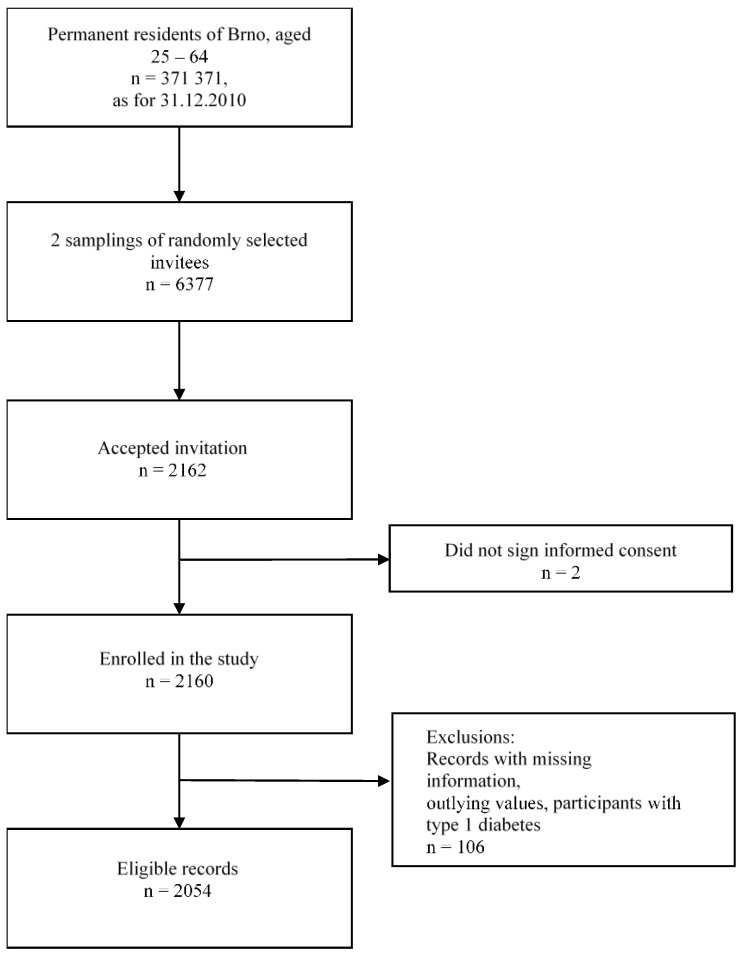
Flowchart of the recruitment, baseline data collection, and selection of the participants for the analysis.

**Table 1 ijerph-18-10251-t001:** Characteristics of the subjects (*n* = 2054).

Variables	Men	Women	*p*
	(*n* = 932)	(*n* = 1122)	
Age categories (%)			
25–34	17.6	15.2	
35–44	26	24.2	
45–54	26.5	26.6	
55–64	29.9	36	0.179
BMI (kg/m^2^)	26.0 (5.0)	24.0 (6.0)	<0.001
Waist Circumference (cm)	95.0 (16.0)	82.0 (18.0)	<0.001
Resting Heart Rate (bpm)	68.6 (14.0)	71.4 (12.2)	<0.001
Hypertension (%)	43.7	33.4	<0.001
Type 2 Diabetes (%)	6.1	3.1	<0.001
Low HDL-c (%)	12.5	14.7	0.158
High LDL-c (%)	69.4	57.1	<0.001
Hypertriglyceridemia (%)	29.5	13.5	<0.001
Hypercholesterolemia (%)	52.2	57.3	0.022
Physically Active (%)	87.7	88.8	0.279
Current Smokers (%)	25.2	22	0.231
Alcohol Users (%)	90	78.8	<0.001
Educational Level (%)			
Low	20.5	18.6	
Middle	33.3	42.6	
High	46.1	38.7	<0.001
Household income (Euro) (%)			
Low (<1200)	33.5	50	
Middle (1200–1800)	34	30.4	
High (>1800)	32.5	19.6	<0.001
Living in Couple (%)	66.4	58.9	<0.001
Medications (%)			
Diuretic	7.6	6.1	0.21
Vasodilator	24.4	19.7	0.011
Hypoglycemic agents	3.5	4.5	0.294
Hypolipidemic agents	11.8	9	0.056

**Table 2 ijerph-18-10251-t002:** Cardiorespiratory fitness classification (METs) according to quartiles of fitness (n = 2054).

Age Categories	Q1 (Lowest)	Q2	Q3	Q4 (Highest)
**Men (*n* = 932)**				
25–34	≤12.0	>12.0–13.3	>13.3–14.0	>14.0
35–44	≤11.9	>11.9–13.0	>13.0–13.9	>13.9
45–54	≤11.0	>11.0–12.3	>12.3–13.0	>13.0
55–64	≤9.9	>9.9–10.9	>10.9–11.9	>11.9
**Women (*n* = 1122)**				
25–34	≤10.2	>10.2–10.6	>10.6–11.0	>11.0
35–44	≤9.6	>9.6–10.3	>10.3–10.8	>10.8
45–54	≤8.8	>8.8–9.6	>9.6–10.2	>10.2
55–64	≤7.9	>7.9–8.6	>8.6–9.2	>9.2

**Table 3 ijerph-18-10251-t003:** Association between CRF (in METs) and presence of cardiometabolic risk factors.

CRF and Hypertension					
Quartiles of Fitness	Hypertension	Model 1 ^a^	95% CI	Model 2 ^b^	95% CI
	(%) **	OR		OR	
Q1—lowest	58.2	1		1	
Q2	40.2	0.48 **	0.38–0.61	0.69 *	0.49–0.96
Q3	28.7	0.29 **	0.22–0.37	0.48 **	0.32–0.72
Q4—highest	21.1	0.19 **	0.14–0.25	0.36 **	0.22–0.60
METs per Unit		0.73 **	0.69–0.77	0.59 **	0.50–0.70
**CRF and Type 2 Diabetes**					
**Quartiles of Fitness**	**Type 2 Diabetes**	**Model 1 ^a^**	**95% CI**	**Model 2 ^c^**	**95% CI**
	**(%) ****	**OR**		**OR**	
Q1—lowest	10.3	1		1	
Q2	4	0.36 **	0.22–0.60	0.48 *	0.26–0.88
Q3	1.6	0.15 **	0.07–0.31	0.22 **	0.09–0.54
Q4—highest	1.1	0.10 **	0.04–0.24	0.16 **	0.05–0.47
METs per Unit		0.79 **	0.71–0.88	0.50 **	0.36–0.70
**CRF and Low HDL-c**					
**Quartiles of Fitness**	**Low HDL-c (%) ****	**Model 1 ^a^**	**95% CI**	**Model 2 ^d^**	**95% CI**
		**OR**		**OR**	
Q1—lowest	26.1	1		1	
Q2	14.9	0.49 **	0.36–0.67	0.86	0.59–1.24
Q3	7.4	0.23 **	0.15–0.33	0.47 **	0.29–0.77
Q4—highest	4.1	0.12 **	0.07–0.20	0.32 **	0.17–0.60
METs per Unit		0.71 **	0.66–0.76	0.82	0.67–1.00
**CRF and High LDL-c**					
**Quartiles of Fitness**	**High LDL-c (%) ****	**Model 1 ^a^**	**95% CI**	**Model 2 ^d^**	**95% CI**
		**OR**		**OR**	
Q1—lowest	66.8	1		1	
Q2	65.4	0.94	0.73–1.20	0.78	0.55–1.10
Q3	61.2	0.78	0.61–1.00	0.53 **	0.36–0.79
Q4—highest	56.1	0.63 **	0.49–0.82	0.33 **	0.21–0.53
METs per Unit		0.90 **	0.86–0.94	0.82 *	0.69–0.97
**CRF and Hypertriglyceridemia**					
**Quartiles of Fitness**	**Hypertriglyceridemia (%) ****	**Model 1 ^a^**	**95% CI**	**Model 2 ^d^**	**95% CI**
		**OR**		**OR**	
Q1—lowest	35.7	1		1	
Q2	23.9	0.57 **	0.44–0.74	0.62 **	0.45–0.86
Q3	14.2	0.30 **	0.22–0.40	0.33 **	0.22–0.50
Q4—highest	5.9	0.11 **	0.07–0.17	0.13 **	0.07–0.23
METs per Unit		0.91 **	0.87–0.96	0.68 **	0.57–0.81
**CRF and Hypercholesterolemia**					
**Quartiles of Fitness**	**Hypercholesterolemia (%) ***	**Model 1 ^a^**	**95% CI**	**Model 2 ^d^**	**95% CI**
		**OR**		**OR**	
Q1—lowest	56.8	1		1	
Q2	58.5	1.07	0.85–1.36	0.89	0.65–1.21
Q3	54.2	0.9	0.71–1.15	0.65 *	0.45–0.62
Q4—highest	49.6	0.75 *	0.58–0.96	0.44 **	0.29–0.69
METs per Unit		0.89 **	0.85–0.93	0.92	0.79–1.06

CRF—cardiorespiratory fitness; T2D—type 2 diabetes; ^a^ unadjusted; ^b^ adjusted by age categories, sex, body mass index, educational level, smoker, alcohol user, and diuretic and vasodilator use; ^c^ adjusted by age categories, sex, body mass index, educational level, smoker, alcohol user, and antidiabetic medication; ^d^ adjusted by age categories, sex, body mass index, educational level, smoker, alcohol user, and lipid-lowering medication; * *p* < 0.05, ** *p* < 0.01. Chi2 for trend across quartiles of fitness.

## Data Availability

The data presented in this study are available upon request from the corresponding author. The data are not publicly available.

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
