# Peer review of "Prediction of Cardiorespiratory Fitness in Czech Adults: Normative Values and Association with Cardiometabolic Health"

_ijerph, 2021, doi:10.3390/ijerph181910251_

Round 1
Reviewer 1 Report
- Authors may wish to add study design and IQR with median age years in an abstract
- The introduction provides a good, generalized background of the topic. The authors have included explanation of the topic and then provide context, and explain what are being challenged or extended to make the introduction more substantial. The authors have introduced the related work clearly.
- Methods section written well. However, authors did not provide information about sample size calculation. I would suggest expanding research design as it is difficult for the reader to search previously published manuscript and review study design.
- The data analysis is quite standard, and looks appropriate for the study.
- The discussion should be of great interest to the readers and offers the “how” and “why” explanations for study findings. It is also a place for authors to consider “glaring” elements of their data and findings. Authors may wish to expand discussion with additional recent relevant studies.
Reviewer 2 Report
Dear Authors,
It was a pleasure to review your manuscript on 'Prediction of Cardiorespiratory Fitness in Czech Adults: Normative Values and Association with Cardiometabolic Health'. There are only a couple of minor considerations before the manuscript can be considered for publication:
- Provide a clear definition of cardiometabolic risk and cardiorespiratory fitness, as used in this study, with reference
- The literature review is limited to this particular population (e.g. check adults, etc), while the literature from other places around the world can give readers a better perspective. Please expand the literature review to include other comparable studies conducted elsewhere together with a critical appraisal to strengthen the justification and rationale of the current study.
- Please clarify how normal distribution was tested, and if all/some variables were meeting the requirements of parametric testing, to justify the statistical analysis conducted.
- Lines 79-82, there are three components defined for the aim. Re-organise and re-present the results with clear reference to how each components of aims have been addressed.
- Please re-organise the abstract and conclusion to provide an outline/take home message of how three aims were addressed.
- Section 2.4, please provide reference for the statements and definitions.
- Lines 218-30, Please critically appraise how/why normative CRF values compare with the ones reported from other populations
Reviewer 3 Report
The authors aim to estimate Cardiorespiratory Fitness of a sample of adults aged between 25 and 64 years old in Czechia. The submitted is wel planned and generally well written. However, the reviewer expects more in depth discussion. It not clear how or why the analyzed variables may have effect on the Cardiorespiratory Fitness in the Discussion section. A detailed discussion may to improve paper and help readers to understand effect of the analyzed variables.
